# Numerical Study of an Ultra-Broadband All-Silicon Terahertz Absorber

**Jinfeng Wang [1], Tingting Lang [1,\*], Tingting Shen [1], Changyu Shen [1], Zhi Hong [2]**  **and Congcong Lu [3]**

[1] Institute of Optoelectronic Technology, China Jiliang University, 258 Xueyuan Street, Hangzhou 310018, China; s1904080311@cjlu.edu.cn (J.W.); s1704080311@cjlu.edu.cn (T.S.); shenchangyu@cjlu.edu.cn (C.S.)

[2] Centre for THz Research, China Jiliang University, 258 Xueyuan Street, Hangzhou 310018, China; hongzhi@cjlu.edu.cn

[3] Hangzhou First Technician College, 719 Xixi Road, Hangzhou 310023, China; cclu@vip.163.com

\* Correspondence: langtingting@cjlu.edu.cn

**Abstract:** In this article we present and numerically investigate a broadband all-silicon terahertz (THz) absorber which consists of a single-layer periodic array of a diamond metamaterial layer placed on a silicon substrate. We simulated the absorption spectra of the absorber under different structural parameters using the commercial software Lumerical FDTD solutions, and analyzed the absorption mechanism from the distribution of the electromagnetic fields. Finally, the absorption for both transverse electric (TE) and transverse magnetic (TM) polarizations under different incident angles from 0 to 70° were investigated. Herein, electric and magnetic resonances are proposed that result in perfect broadband absorption. When the absorber meets the impedance matching principle in accordance with the loss mechanism, it can achieve a nearly perfect absorption response. The diamond absorber exhibits an absorption of ~100% at 1 THz and achieves an absorption efficiency >90% within a bandwidth of 1.3 THz. In addition, owing to the highly structural symmetry, the absorber has a polarization-independent characteristic. Compared with previous metal–dielectric–metal sandwiched absorbers, the all-silicon metamaterial absorbers can avoid the disadvantages of high ohmic losses, low melting points, and high thermal conductivity of the metal, which ensure a promising future for optical applications, including sensors, modulators, and photoelectric detection devices.

**Keywords:** broadband absorption; all-silicon terahertz absorber; electric and magnetic resonances

## 1. Introduction

Terahertz (THz) waves refer to the electromagnetic waves whose frequency ranges are between 0.1 and 10 THz [1]. Owing to the unique properties of THz radiation, including its capacity for increased penetration compared to visible light in many dielectrics [1], low photon energy [2], and high spectral-resolving ability, it is extensively used in various fields, such as in THz imaging, explosives detection, communication, food quality control, and others [3–6]. Metamaterials are a new kind of artificial material, which are cell structures periodically arranged by subwavelength dimension [7,8]. Metamaterial-based absorbers can absorb incident light efficiently and convert it into ohmic heat or other forms of energy. Because only a few substances in nature respond to THz frequencies [9], it is possible to achieve efficient reactions with artificial metamaterials by designing the structural units. Silicon is a common semiconductor and has many advantages, including its high electron mobility, easy integration, and low cost. In 2008, Landy et al. designed a "perfect" metamaterial absorber that achieved nearly 100% absorption in the GHz frequency band. Later in the same year, Hu Tao et al. proposed an absorber with a metal ring resonator and a split wire separated by a dielectric, which

was supported by gallium arsenide (GaAs). This proposed absorber, which operated in the THz band on the basis of Landy's findings, achieved a nearly perfect absorption in this band for the first time, and reached a structure at the micron scale [10]. In 2009, Landy et al. designed an absorber with a modified electrically coupled ring resonator with a cross structure that realized ~95% absorption at 1.13 THz [11]. The proposed absorber created new research opportunities in the research of metamaterial absorbers [12–14]. These characteristics led to the research boom of the THz absorber [10]. In subsequent studies, various schemes were proposed for perfect absorption based on metamaterials with operations which ranged from the microwave to THz [11,15,16], infrared [17–20], and in visible wavelength bands [21,22]. In order to achieve perfect absorption in the visible and infrared wave bands, the biggest challenge was to make the size of the unit structure smaller than the wavelength of the incident electromagnetic waves. Nanometer lithography technology can realize the fabrication of devices, but it is difficult to guarantee the machining accuracy. In 2017, Jianhao Gong et al. proposed 2D all-metal gradient nanostructures that could realize >90% absorption over a wide wavelength band of 378–626 nm [23]. This all-medium absorber greatly simplified the manufacturing process. In the same year, Dong Wu et al. designed a metal stripe placed on a thin $SiO_2$ layer, which was supported by the metal–dielectric layer. This absorber, with metal resonator, could retain absorption above 95% at the wavelength range of 400 to 1500 nm [24]. In 2018, Lei Lei et al. proposed an ultra-broadband absorber based on a metamaterial nanostructure composed of a periodic array of titanium–silica (Ti–$SiO_2$) cubes and aluminum (Al) bottom films that showed an average absorption of 97% from 354 to 1066 nm [25].

Traditional absorbers based on metal–dielectric–metal sandwiched structures mainly achieve single-frequency and multiple-band absorption. Because of the complexity of this multilayered structure, the manufacturing process is complicated, and the cost of fabrication is expensive. Recently, all-dielectric broadband THz metamaterial absorbers have become a research hotspot [26]. In 2014, Withawat Withayachumnankul et al. designed an all-silicon circular groove absorber that achieved a bandwidth of 360 GHz above 90% absorbance [27]. In 2018, Xiaoguang Zhao et al. proposed an all-silicon H-shaped microstructure with two rectangular cavities on each sidebar based on a substrate that achieved the peak absorption absorbance ~99.99%, with a bandwidth of ~900 GHz for 90% absorbance [26]. In 2019, Jianwen Xie et al. designed a dielectric water-based metamaterial absorber that realized absorption above 90% from 8 to 20 GHz [28].

In this study, the perfect absorption mechanism is explained based on electric and magnetic resonances excited in the diamond metamaterial layer, based on free-space impedance that results in the dissipation of large amount of energy at resonance. The resonant frequencies are related to the sizes of the structure and can be tuned by adjusting the length and the thickness of the diamond layer. More importantly, even though the resonant frequency changes, the absorption remains high. In addition, our proposed structure has the characteristics of polarization independence and broadband absorption. In the case of transverse electric (TE) polarization, the absorber can remain in a high-absorption state from 0 to 70°, but in the case of transverse magnetic (TM) polarization, it can only remain in the range of 0 to 40°. Compared with other absorbers listed in Table 1 with the structure of $VO_2$-$SiO_2$-Au [29], cross-shaped graphene-$SiO_2$-Au [30], graphene disks-$SiO_2$ [31], all-silicon V-groove [32] et al, the all-silicon diamond absorber can also achieve a broad absorption band with an absorption efficiency >90%.

**Table 1.** Comparison with other absorbers.

|  | References | Structure | Waveband (THz) | Peak Absorption | Above 90% |
|---|---|---|---|---|---|
| **Dielectric–metal** | [1] | Cu-SiO$_2$-Si | 0.2–2 | 98% | 0.08 THz |
|  | [4] | Continuous graphene-SiO$_2$-Au | 0–2 | 98% | 1 THz |
|  | [29] | VO$_2$-SiO$_2$-Au | 0.2–2 | Nearly 100% | 0.7 THz |
|  | [30] | Cross-shaped graphene-SiO$_2$-Au | 1–8 | Nearly 100% | 1.13 THz |
| **All-dielectric** | [31] | Graphene disks-SiO$_2$ | 0–5 | Nearly 100% | 0.4 THz |
|  | [27] | All-silicon circular groove | 0–2 | Nearly 100% | 0.36 THz |
|  | [32] | All-silicon V-groove | 0.5–2 | 99% | 0.4 THz |
|  | [33] | All-silicon hypersurface | 0.2–1.5 | 93.80% | 0.05 THz |
|  | This study | Diamond | 0.2–2 | Nearly 100% | 1.3 THz |

## 2. Structure and Design

Figure 1 shows the proposed ultra-broadband THz metamaterial absorber with a magnified cubic unit cell. We can observe that the absorber consists of all-silicon diamond metamaterial layers placed on the base of the cubic substrate. The height of the metamaterial layer and the structural base are denoted as $t$ and $h$, respectively, while the period of the absorber is denoted as $p$ and the length of the diamond is $a$. These parameters are optimized as follows: $p = 170$ μm, $h = 250$ μm, $t = 60$ μm, $a = 75\sqrt{2}$ μm. The FDTD Solutions software produced by the Canadian Lumerical Solutions company (Vancouver, BC, Canada) was used to obtain the reflectance (R) and transmittance (T) of the structure at different geometrical parameters. This software adopts the finite difference time domain (FDTD) method. This algorithm uses Maxwell equations to analyze the interaction between electromagnetic waves and structures with subwavelength dimensions. Subsequently, the absorption (A) was calculated using the equation $A = 1 - R - T$.

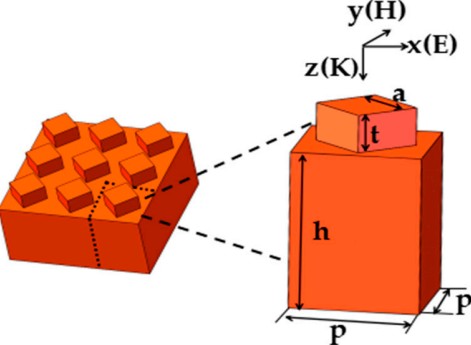

**Figure 1.** Schematic of the proposed terahertz metamaterial absorber.

We can design and explain the working principle for the investigated structures. The silicon used in this study is modeled by the Drude response model [1]:

$$\varepsilon(w) = \varepsilon_\infty - \frac{w_p{}^2}{w^2 + jw\gamma} \tag{1}$$

in which $\varepsilon_\infty = 11.68$ is the intrinsic permittivity of Si, and $f$ is the electromagnetic frequency. The plasma frequency $W_P$ and the damping rate $\gamma$ are related with the free carrier concentration $N$ and the free carrier mobility $\mu$, respectively. The equations follow

$$w_p{}^2 = \frac{Ne^2}{\varepsilon_0 m^*} \tag{2}$$

$$\gamma = \frac{e}{\mu m^*} \tag{3}$$

$$\delta = Ne\mu \tag{4}$$

where $e$ is the electron charge, $m^*$ is the effective mass of free carriers and has a value of 0.37 $m_0$, $m_0$ is the mass of the electron, $\delta$ is the conductivity, and $\varepsilon_0$ is the permittivity of free space. The empirical parameter of $\mu$ is chosen to have a value of 386 cm$^2$ V$^{-1}$ S$^{-1}$. Its carrier density was derived from the conductivity of silicon wafers available on the market, which is approximately equal to $0.03 \times 10^{18}$ cm$^{-3}$. Finally, we calculated the plasma frequency value of $1.6042 \times 10^{13}$ rad/s and the collision frequency value of $1.9573 \times 10^{12}$ 1/s.

To gain a better understanding of the absorption performance, we also analyzed the impedance matching mechanism. Herein, an effective medium theory (EMT) is proposed. The absorber proposed in this study was considered to be homogeneous. The $S$-parameters and impedance $Z$ can be expressed as [24,34]

$$S_{21} = \frac{1}{\cos(nkd) - \frac{i}{2}(Z + \frac{1}{Z})\sin(nkd)} \tag{5}$$

$$S_{11} = \frac{i}{2}(\frac{1}{Z} - Z)\sin(nkd) \tag{6}$$

where $S_{21}$ is the coefficient of transmission, $S_{11}$ is the coefficient of reflection, $n$ is the effective refractive index of the absorber, $k$ is the wave vector, and $d$ is the thickness of the absorber. The impedance $Z$ and absorption $A$ are then obtained:

$$Z = \pm\sqrt{\frac{(1 + S_{11})^2 - S_{21}{}^2}{(1 - S_{11})^2 - S_{21}{}^2}} \tag{7}$$

$$A = 1 - S_{11}{}^2 - S_{21}{}^2 \tag{8}$$

The impedance $Z_0$ of free space is equal to unity. When the impedance matches the free-space impedance ($Z = Z_0 = 1$) at the working frequency, a nearly perfect absorption can be obtained. Based on calculations, the impedance at the two resonant peaks is closer to unity.

## 3. Simulated Results and Discussion

First, the polarization characteristic was analyzed, as shown in Figure 2. For TE polarization, the electric field is transverse along the x-axis, and for TM polarization, the magnetic field is also transverse along the x-axis. It can be observed that the spectral curves of the two polarizations are highly coincident owing to the highly structural symmetry. As observed from the figure shown above, absorption rates of nearly 100% and 97.8% can be achieved at the resonant frequencies of 1 and 1.72 THz, respectively. The all-silicon terahertz absorber has an absorption bandwidth of 1.3 THz with an absorption efficiency >90%. We can conclude that it has a polarization independence characteristic and a broad absorption bandwidth.

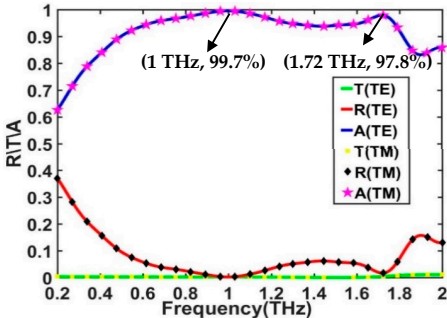

**Figure 2.** Reflection, transmission, and absorption spectral outcomes of transverse electric (TE) and transverse magnetic (TM) polarizations.

To gain a better understanding of this absorption effect, we also investigated the electromagnetic field distributions at the two resonant peaks in the presence of TE polarization with a normal incidence. Figure 3a,b,e,f respectively show the electric and magnetic field distributions on the upper surface of the silicon layer at the resonant frequencies of 1 and 1.72 THz. Figure 3c,g respectively show the electric field distribution at the x-z cross-section when y = 10 μm at the two resonant frequencies. Figure 3d,h respectively show the magnetic field distribution at the y–z cross-section when x = 10 μm at the two resonant frequencies. Figure 3a,c show that the electric field is mainly concentrated in the diamond metamaterial edges and the air-slot between the adjacent unit cells at the resonant frequency of 1 THz, thus indicating the occurrence of the electric resonance [34]. Additionally, Figure 3b,d show that the magnetic field is mainly located in the diamond layer. This indicates that the magnetic resonance is also excited [34]. However, as shown in Figure 3f,h, the magnetic field intensity increases considerably at the resonant frequency of 1.72 THz compared with the first resonant peak. This indicates that the magnetic resonance is significantly enhanced. It is worth noting that the electric field is stronger than the magnetic field, so both absorption peaks are dominated by electric resonance. As shown in Figure 3a,c,e,g, the strength of the electrical resonance of peak 1 is higher than that of the peak 2, so the absorption is higher. In summary, the appearances of the electric and magnetic resonances lead to an almost perfect broadband absorption.

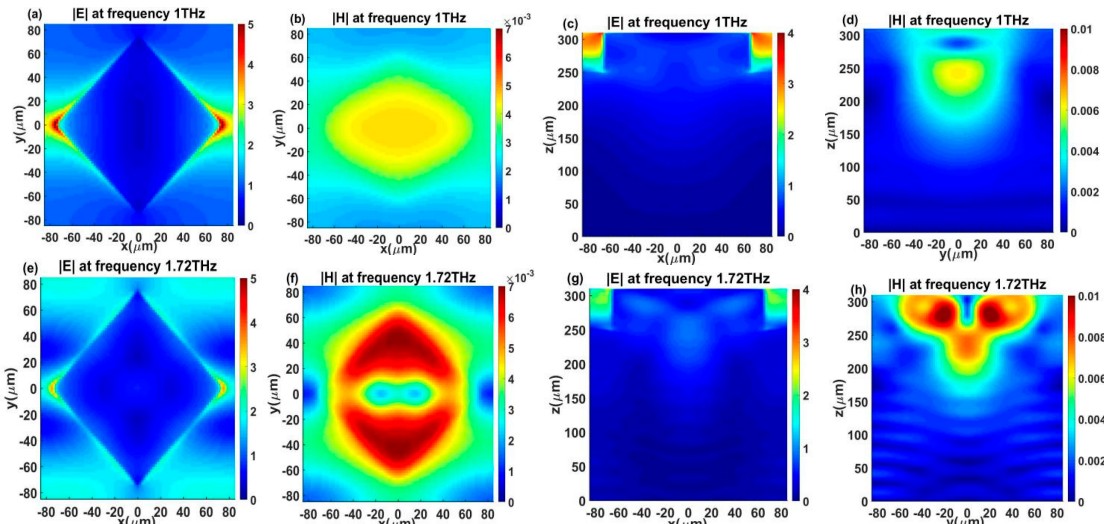

**Figure 3.** (**a**,**b**), and (**e**,**f**), respectively, indicate the electric and magnetic field intensities on the upper surface of the silicon layer at the two resonant peaks. (**c**,**g**) show the electric field intensity distributions at the x–z cross-section when y = 10 μm, while (**d**,**h**) indicate the magnetic field intensity distributions at the y–z cross-section when x = 10 μm.

We also analyzed the influence of the geometrical shapes on the absorption properties. First, we analyzed the effect of the diamond metamaterial layer on the absorption characteristic. The simulated results of the absorber for a normal incidence for TE polarization using only the substrate (red) and the diamond absorber (black) are shown in Figure 4a. The size of the Si substrate has not changed. It can be observed that the two resonant peaks and the bandwidth for 90% absorbance disappear in the absence of a diamond metamaterial layer. In order to reflect the absorption effect of the diamond metamaterial layer, we also analyzed the absorption effect of a non-patterned (continuous) top layer placed on the substrate, with the thickness values *b* of 40, 50, 60, 70, and 80 μm, respectively. As shown in Figure 4b, it can be clearly seen that increasing the thickness of the top layer has no effect on the absorption efficiency, and the absorption is less than 80%. This may be because increasing the thickness of the top layer is equal to increasing the thickness of the substrate. Therefore, compared with Figure 4a, it can be concluded that the diamond metamaterial layer enhances the absorption.

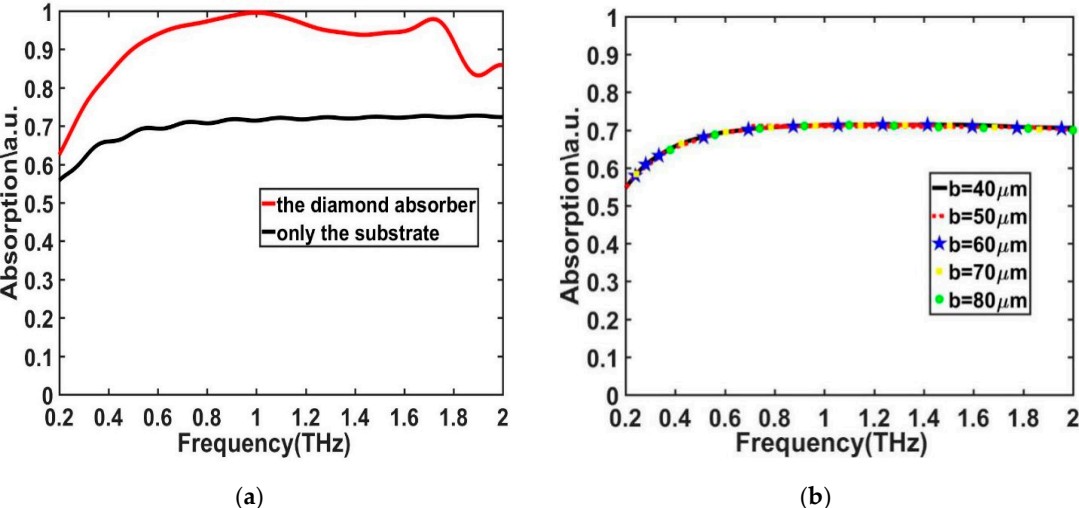

**Figure 4.** (**a**) Absorption spectra of only the silicon substrate compared with the diamond absorber. (**b**) Absorption spectra in the case of TE polarization for five different thickness values *b* of a non-patterned (continuous) top layer placed on the substrate.

Next, we adjusted the thickness of the diamond metamaterial layer, but all the other parameters remained unchanged. Figure 5a presents the numerically simulated absorption spectrum for different thickness values *t*. We can observe that each absorption spectrum has two resonance peaks, one at the lower frequency and the other at a higher frequency. The two resonant peaks undergo a right shift as the thickness of the diamond layer decreases, while the peak absorption is almost unchanged. As shown in Figure 3c,g, the change of the thickness of the diamond layer has a more profound influence on the electric field at the first resonant peak. Thus, peak 1 exhibits a significant deviation. Therefore, we can design the thickness of the metamaterial layer according to the actual needs to obtain increased absorption responses in specific bands.

We then discuss the length of the diamond diagonal on the absorption efficiency where the period *p* remains unchanged at 170 μm. When the lengths of the diagonal are 170, 150, 130, and 110 μm, this is the same as when the sides *a* of the diamond are 85 $\sqrt{2}$, 75 $\sqrt{2}$, 65 $\sqrt{2}$, and 55 $\sqrt{2}$ μm. As shown in Figure 5b, when the length of the diamond decreases from 85 $\sqrt{2}$ to 55 $\sqrt{2}$ μm, the first absorption peak is gradually increased and then decreased, while the second absorption peak has no obvious change. This is because changing the length of the diamond has a greater effect on the electric field at peak 1, as shown in Figure 3c,g, so peak 1 has a greater change. It is worth noting that when the length of the diamond is 75 $\sqrt{2}$ μm, the first absorption peak reaches nearly perfect absorption. This is because the electric field is localized better, thus achieving the enhancement of electric resonance.

Also, we plotted the absorption spectrum in Figure 5c to assess the influence of the value of the periodic *p*. As shown in Figure 5c, when we increase the length from 170 to 200 μm, and with all the other parameters unchanged, there is no significant change in the first resonant peak. By contrast, the second peak shifts significantly to the left and yields an absorption efficiency >97%. Because the change of the periodic has a considerable effect on the magnetic field of peak 2, as shown in Figure 3b,f, the second resonant peak exhibits an obvious deviation.

Finally, we discuss the effect of the thickness of the substrate on absorption, with the thickness values *h* of 150, 200, 250, and 300 μm, respectively. It can be clearly seen from Figure 5d that when the thickness of the substrate is reduced from 300 to 150 μm, there is no significant effect on the absorption. This is because when the substrate is thick enough to block the transmission of electromagnetic waves, there is no significant effect on absorption. In order to achieve maximum absorption in the THz regions, the optimized structural parameters of the all-silicon diamond absorber are *p* = 170 μm, *h* = 250 μm, *t* = 60 μm, and *a* = 75 $\sqrt{2}$ μm. The structural parameters of the absorber can be reasonably designed to obtain a specific absorption broadband range.

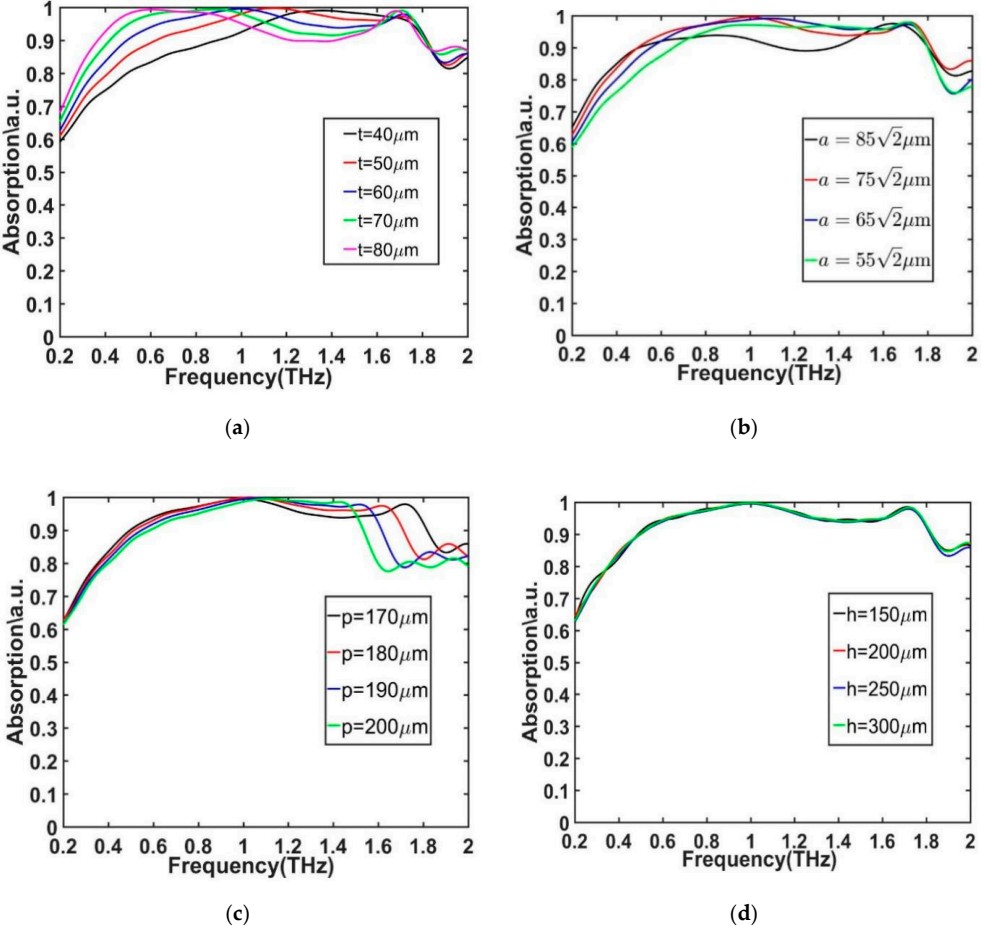

**Figure 5.** Absorption spectra of the diamond absorber in the case of TE polarization for five different thickness values *t* of the diamond metamaterial layer (**a**), for four different lengths values *a* of the diamond layer (**b**), for four different lengths values *p* of periodic (**c**), and for four different thickness values *h* of the substrate (**d**).

The aforementioned analysis refers to the absorption characteristic of incident light at a normal angle. In practice, the light source is incident at an oblique angle on the device, and it is thus valuable to design an absorber with a strong absorption at a large angle. To obtain the absorber with high-absorption performance, it is necessary to discuss the influences of the incident angles for both the TE and TM polarizations. At non-zero angles of incidences, we chose the plane wave related with the broadband fixed angle source technique (BFAST). Figure 6a,b show the contrast of light absorption at different incident angles. In the case of TE polarization, the first resonant peak maintains an almost perfect absorption response at 45°, while the second resonant peak has a redshift as the incident angle increases. It is worth noting that the absorption bandwidth above 90% is maintained up to 70°. This is owing to the fact that the magnetic field is maintained when the incident angle is varied. Accordingly, the magnetic resonant strength can be enhanced. In the case of TM polarization, the absorption efficiency is weakened when the incident angles are greater than 40°. The second absorption peak is less than 90% when the incident angle is greater than 10°. This is because that the direction of the magnetic field varies with the angle of incidence, which results in the decrease of magnetic resonant strength. This all-silicon terahertz absorber can maintain wide band absorption under large angle incidence, so it can have great application value.

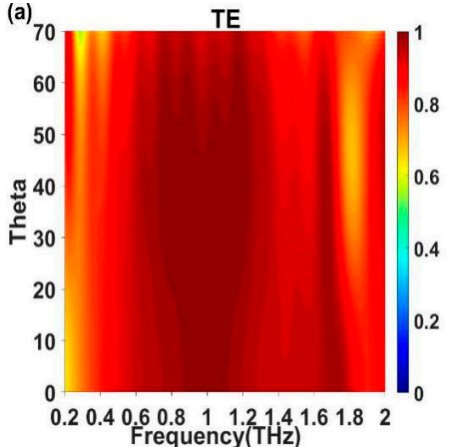 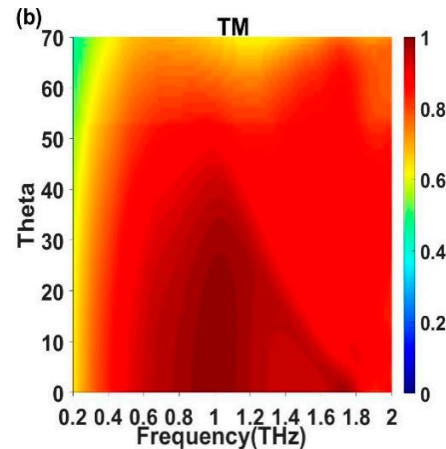

**Figure 6.** Absorption spectrum at different incident angles for (**a**) TE polarization and (**b**) polarization.

## 4. Conclusions

In summary, we proposed and numerically investigated an ultra-broadband all-silicon THz absorber with a microstructure configured by a periodic array of diamond metamaterial layer and a cubic substrate using the FDTD method. This type of absorber exhibited a nearly perfect absorption response, and had a very wide absorption bandwidth of ~1.3 THz for absorption efficiency of >90%. The electric and magnetic resonances of the silicon layer enabled this nearly perfect broadband absorption characteristic. The advantages of our perfect absorber included polarization independence, wideband absorption, angle insensitivity, and a simple configuration. Moreover, compared with previous metal–dielectric–metal absorbers, the all-dielectric absorber has numerous excellent characteristics. Additionally, the silicon costs less and is less difficult to process, making it possible to apply it in various fields. In future work, we plan to use the absorber for THz imaging and detection.

**Author Contributions:** J.W. wrote the article; T.L. reviewed and edited; T.S. validated, C.S. analyzed the simulations; Z.H. put forward some comments; C.L. checked the spelling, grammar of the article and investigated. All authors have read and agreed to the published version of the manuscript.

**Funding:** This work was supported in part by the National Natural Science Foundation of China grant number 61875251, 61875179 and 11874332; in part by the Public project of Zhejiang Province grant number LGG18F050003 and in part by the National major scientific research instrument development project of Natural Science Foundation of China grant number 61727816.

**Conflicts of Interest:** The authors declare no conflicts of interest.

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
