# Peer review of "Numerical Study of an Ultra-Broadband All-Silicon Terahertz Absorber"

_applsci, doi:10.3390/app10020436_

Round 1

Reviewer 1 Report

Manuscript No:  applsci-670189

Title:  Ultrabroadband all-silicon terahertz absorber

Authors:  Jinfeng Wang,Tingting Lang1,Tingting Shen1, Changyu Shen1, Zhi Hong and Congcong Lu

Overview In this manuscript the authors report on the simulation of a broadband all-silicon terahertz absorber

consisting of a single-layer array of a metamaterial placed on a silicon substrate.

The contents are expressed clearly, the manuscript is well organized and written in reasonable English. Look for typos. The authors have acknowledged recent related research. As long as my knowledge, the work presented is original and it is correct from a scientific point of view.

Detailed analysis

Title: please change it: it must inform readers that this is a simulation work.

Abstract: Must be clear and objective.

Please state what have you done, how did you do it, the results you got and the novelty of your work

Introduction: provides the relevant information and up to date references.

Structure and design

How did you implement the finite difference time domain (FDTD) algorithm?

Details must be present on the algorithm/method and on the software platform utilized (commercial, homemade?).

A section should not start with a figure!

Always describe and comment a figure before the reader see it in the text.

How do you optimized the parameters: p = 170 μm, h = 250 μm, t = 60 μm, a = 85 2 μm?

I only see optimization of parameter t and p in section 3.

Please provide a comprehensive discussion on the selection of the parameters.

In addition, simulation must be carried out and presented to achieves these 4 parameters.

Simulated results and discuss

Equations (5) to (8) should be introduced in section 2.

Figure 4 and 5: units of Absorption?

Overall assessment

The work reported presents reasonable utility for supplementary studies and developments in the field.

In my opinion it may be eventually published after major revision.

Review Criteria Scope of Journal

Rating: Moderately high

Novelty and Impact

Rating: Medium

Technical Content

Rating: Medium

Presentation Quality

Rating: Medium

Reviewer 2 Report

The authors present a diamond patterned silicon layer which enhances THz absorption. I have a few questions comments for the authors which could be addressed to improve the quality of their manuscript.

1. The resolution of all figures should be increased if possible.

2. Figure 2 - the colors are very difficult to make out what is what. A more traditional red, green, blue color scheme for A, T, R, respectively, would help here. Also, making the linewidths thicker would help improve the quality.

3. Based on the electric field plots shown in Figure 3, it doesn't seem like the diamond layer localizes the fields in the substrate very well. Also, given the substrate's large absorption it would be nice to know how much the diamond layer enhances the absorption. The authors could simulate the substrate with a non-patterned (continuous) top layer of the same t=40,50, 60,70,80 um thicknesses and compare that to the data shown in Figure 4/5. 

4. Also, the authors are encouraged to greatly expand the reference list, particularly for the background on metamaterial absorbers. The Landy "perfect metamaterial absorber" should be cited. Additionally, there are a large number of infrared metamaterial absorber papers the authors could cite. 

Round 2

Reviewer 1 Report

Manuscript No:  applsci-670189 R1

Title:  Numerical study of an ultra-broadband all-silicon terahertz absorber

Authors:  Jinfeng Wang,Tingting Lang,Tingting Shen, Changyu Shen, Zhi Hong and Congcong Lu

A. Overview

1. In this manuscript the authors report on the simulation of a broadband all-silicon terahertz absorber

consisting of a single-layer array of a metamaterial placed on a silicon substrate.

2. The contents are expressed clearly; the manuscript is well organized and written in reasonable English.

3. The authors have acknowledged recent related research.

4. As long as my knowledge, the work presented is original and it is correct from a scientific point of view.

B. Overall assessment

The work reported presents reasonable utility for supplementary studies and developments in the field and it is likely to have an impact on journal readers.

The authors have introduced corrections to all the comments and answered to the queries of the reviewers.

In my opinion it may be published as is.

C. Review Criteria

1. Scope of Journal

Rating: Moderately high

2. Novelty and Impact

Rating: Medium

3. Technical Content

Rating: Medium

4. Presentation Quality

Rating: Medium